# Can the Brazilian Caffeine Expectancy Questionnaires Differentiate the *CYP1A2* and *ADORA2A* Gene Polymorphisms?—An Exploratory Study with Brazilian Athletes

**DOI:** 10.3390/nu14163355

**Published:** 2022-08-16

**Authors:** Guilherme Falcão Mendes, Caio Eduardo Gonçalves Reis, Eduardo Yoshio Nakano, Higor Spineli, Gabriel Barreto, Natália Yumi Noronha, Lígia Moriguchi Watanabe, Carla Barbosa Nonino, Gustavo Gomes de Araujo, Bryan Saunders, Renata Puppin Zandonadi

**Affiliations:** 1Department of Nutrition, School of Health Sciences, University of Brasilia (UnB), Campus Darcy Ribeiro, Asa Norte, Brasilia 70910-900, DF, Brazil; 2Department of Statistics, Central Institute of Sciences, University of Brasilia (UnB), Campus Darcy Ribeiro, Asa Norte, Brasilia 70910-900, DF, Brazil; 3Research Group Applied to Sport Science—GPCAE, Institute of Physical Education and Sport—IEFE, Federal University of Alagoas, Maceió 57072-970, AL, Brazil; 4Applied Physiology and Nutrition Research Group, School of Physical Education and Sport, Faculty of Medicine FMUSP, University of São Paulo, São Paulo 05405-000, SP, Brazil; 5Department of Internal Medicine, Ribeirão Preto Medical School, Ribeirão Preto 14049-900, SP, Brazil; 6Department of Health Sciences, Ribeirão Preto Medical School, Ribeirão Preto 14049-900, SP, Brazil; 7Rheumatology Division, Faculty of Medicine FMUSP, University of São Paulo, São Paulo 05405-000, SP, Brazil

**Keywords:** caffeine, *CYP1A2*, *ADORA2A*, polymorphism, expectancy questionnaire

## Abstract

This study investigated the ability of the Brazilian Caffeine Expectancy Questionnaire (CaffEQ-BR), full and brief versions, to differentiate genetic profiles regarding the polymorphisms of the *CYP1A2* (rs 762551) and *ADORA2A* (rs 5751876) genes in a cohort of Brazilian athletes. One-hundred and fifty participants were genotyped for *CYP1A2* and *ADORA2A*. After the recruitment and selection phase, 71 (90% male and 10% female, regular caffeine consumers) completed the CaffEQ-BR questionnaires and a self-report online questionnaire concerning sociodemographic data, general health status, and frequency of caffeine consumption. The order of completion of the CaffEQ-BR questionnaires was counterbalanced. The concordance between the full and brief versions of the CaffEQ-BR was analyzed using the intraclass correlation coefficient (ICC). To determine the discriminatory capacity of the questionnaires for genotype, the receiver operating characteristic (ROC) curve was applied for sensitivity and specificity (significance level of 5%). Mean caffeine intake was 244 ± 161 mg·day^−1^. The frequency of AA genotypes for *CYP1A2* was 47.9% (*n* = 34) and 52.1% (*n* = 37) for C-allele carriers (AC and CC). The frequencies of TT genotypes for *ADORA2A* were 22.7% (*n* = 15) and 77.3% (*n* = 51) for C-allele carriers (TC and CC). All CaffEQ-BR factors, for the full and brief versions, were ICCs > 0.75, except for factor 6 (anxiety/negative effects; ICC = 0.60), and presented ROC curve values from 0.464 to 0.624 and 0.443 to 0.575 for CYP1A2 and ADORA2A. Overall, the CaffEQ-BR (full and brief versions) did not show discriminatory capacity for *CYP1A2* and *ADORA2A* gene polymorphisms. In conclusion, the CaffEQ-BR was not able to differentiate genotypes for the *CYP1A2* or *ADORA2A* genes in this group of Brazilian athletes.

## 1. Introduction

Caffeine is the most consumed psychoactive substance worldwide, and coffee is the most common source on a global scale [1]. In addition, caffeine can be found in guarana, yerba mate, green tea, cocoa and its derivatives, cola-based beverages, supplements (e.g., energy drinks and chewing gum), and medicines [2,3]. To improve strength, power, and endurance sports performance, the recommended dose varies from 3 to 6 mg·kg^−1^ body mass (BM) [4,5,6]. To improve cognitive aspects and wakefulness, 1 to 3 mg·kg^−1^ BM is an effective dose [7,8]. It is recommended that individuals who have never used caffeine supplementation or those who are sensitive to caffeine use lower doses [9]. Furthermore, high doses can be associated with increased adverse effects, such as irritation, anxiety, tachycardia, and sleep disturbance, depending on the genetic profile of the individual [10].

Genetic variants can affect caffeine metabolism [11], potentially inducing different effects of caffeine regarding the perception of effort, fatigue, sleep, appetite, and adverse or beneficial effects across a range of exercise modalities, including endurance, team sports, short-duration high-intensity exercise, and resistance exercise [12,13]. In nutrigenetic studies, caffeine is one of the most studied substances in clinical trials assessing the interactions between gene polymorphisms and sports performance [14,15,16,17]. Differences in the effect of caffeine may be explained in part by the polymorphism in the single nucleotide polymorphism (SNP) of the cytochrome P450 enzyme *CYP1A2* (SNP rs762551 −163C > A; AA and C-carriers genotypes), which is related to the hepatic metabolization of caffeine [11]. C allele carriers (AC and CC genotypes) are considered slow caffeine metabolizers and AA genotypes are fast metabolizers [10]. In addition, a SNP on the *ADORA2A* gene (SNP rs5751876 1976T > C; TT and C-carriers genotypes) is related to the sensitivity and responsiveness to caffeine [18]. TT homozygotes experience high sensitivity to the effects of caffeine, likely due to a greater effect of caffeine on the central nervous system (CNS) adenosine receptors, whereas C allele carriers (TC and CC genotypes) present lower caffeine sensitivity [10]. This effect may lead to feelings of anxiety, irritation, tachycardia, and sleep disturbance in TT homozygotes [19]. 

Understanding how the *CYP1A2* and *ADORA2A* genotypes affect the responsiveness and sensitivity of caffeine in the CNS is of great scientific and clinical relevance [12], including for sports performance, to individualize caffeine supplementation [10,14,20,21]. In clinical practice, this knowledge is important to avoid prescriptions disregarding the patient’s genotype characteristics, or to prevent unnecessary use or sub-optimal caffeine doses according to the patient’s profile [22,23]. In clinical trials, it is important to understand the caffeine expectation profile to minimize the risk of bias related to the individual’s expectancy of caffeine’s effects on exercise performance [7,24]. Thus, it is of interest to determine the *CYP1A2* and *ADORA2A* genotypes of any individuals undergoing caffeine consumption, be it in research or in the real world, since these genes may alter responses. Nonetheless, genotyping is a time consuming and expensive process that is not easily available, meaning alternative, more cost-effective methods are desirable.

Therefore, assessing the expectations about caffeine’s effects is important due to its impacts on mood and sports performance. The Brazilian caffeine expectancy questionnaires (full and brief versions) (CaffEQ-BR) were developed to investigate expectations relating to caffeine ingestion in the adult Brazilian population, specifically regarding factors related to dependence, energy/work improvement, social/mood enhancement, physical performance, anxiety, and sleep disturbances [25,26]. Understanding these expectations can be a useful tool to assess the potential ergogenic or ergolytic response to caffeine supplementation. Physical performance may be differentially affected by caffeine depending on *CYP1A2* genotype [15], whereas anxiety and sleep disturbance have been associated with the *ADORA2A* gene [27]. Thus, genetic differences in caffeine metabolism (e.g., *CYP1A2*) and sensitivity (e.g., *ADORA2A*) may lead to different response profiles to the CaffEQ-BR questionnaires.

The present study aimed to examine the ability of the CaffEQ-BR questionnaires (full and brief versions) to differentiate the polymorphisms of the *CYP1A2* and *ADORA2A* genes. The study hypothesis was that the CaffEQ-BR instrument could differentiate the polymorphisms of the *CYP1A2* and *ADORA2A* genes. If this holds true, these questionnaires may be a cost-efficient method capable of identifying different genotypes that could be implemented in clinical research and practice to guide caffeine recommendations.

## 2. Materials and Methods

This was a cross-sectional study involving 71 participants genotyped for *CYP1A2* and *ADORA2A* polymorphisms who completed the full and brief versions of the CaffEQ-BR [25,26]. The present study was approved by the Ethics Committee of the Universidade Católica de Brasília (CAAE 23019319.3.0000.0029) and followed the guidelines established by the Declaration of Helsinki. Volunteers were informed about the study protocol and provided their consent using an online form. 

### 2.1. Participants

Participants were recruited among individuals who participated in the study by Spineli et al. [28] (*n* = 100) and Barreto et al. (*n* = 50) (data not yet published). Inclusion criteria were: (i) Brazilian adult (19–59 years old), resident in Brazil, regular consumer of caffeine (≥3x a week) from various sources, and agreement to complete the full and brief versions of CaffEQ-BR; (ii) had *CYP1A2* −163C > A and *ADORA2A* 1976T > C genotypes determined. 

In the Spineli et al. study [28], the participants were healthy, trained/developmental athletes [29] engaged in volleyball, athletics, or competitive soccer (age: 15 ± 2 years; height: 1.69 ± 0.10 m; BM: 58.8 ± 11.9 kg; VO_2max_: 44.0 ± 2.7 mL·kg^−1^·min^−1^) [28]; and in Barreto et al. (unpublished), participants were healthy male and female trained cyclists [30] (age: 37 ± 6 and 40 ± 2 y; height: 1.76 ± 0.04 and 1.63 ± 0.04 m; BM: 74.1 ± 6.7 and 61.5 ± 7.3 kg; VO_2max_: 51.1 ± 5.2 and 42.3 ± 8.12 mL·kg^−1^·min^−1^, respectively). It is important to emphasize that the athletes in the Spineli et al. study [28] were teenagers during the collection period of the original study. However, the application of the CaffEQ-BR occurred four years later, and all volunteers were over 19 years old when completing the questionnaires.

The distribution across genotypes generally includes fewer homozygous participants carrying the CC genotype for *CYP1A2* and TT for *ADORA2A*, which represent less than 10% and 20% of the population, respectively [15,27]. Thus, the objectives were to obtain a sample where each allele subgroup had at least 10 genotyped participants. Considering the formation of four subgroups with at least 10 carriers of the C allele (AC and CC) and A homozygote polymorphism (AA) for *CYP1A2* and carriers of the C allele (CT and TT) and T homozygote (TT) *ADORA2A*, the minimum sample would be 40 participants. Of the 150 individuals invited, 71 participated in the research.

### 2.2. Questionnaires Application

Participants completed the full and brief versions of the CaffEQ-BR questionnaire. We have previously shown the full (overall ICCs > 0.9) [25] and brief (overall ICCs > 0.9) [26] versions of the questionnaires to have excellent reliability in the Brazilian population. The questionnaires were applied via Google Forms™ (Google LLC, Mountain View, CA, US) to a convenience sample of adult Brazilian athletes [25,26]. Participants were contacted via phone calls, email, or social media like Facebook™, Instagram™, or WhatsApp™ (Meta Inc., Menlo Park, CA, US) [31]. The data collection period took place between October 2021 and April 2022. First, the participants completed a self-report online questionnaire concerning sociodemographic data, general health status, and frequency of caffeine consumption. Then, they answered the full and brief version of the CaffEQ-BR with a minimum interval of 48 h and a maximum of 15 days between the first and second questionnaires [32]. This step was used to analyze the agreement between the full and short versions of the CaffEQ-BR. The order of completion of the questionnaires was counterbalanced, in which part of the sample started with the full version and then the brief version, and the other part was performed in reverse order [26]. 

### 2.3. Genetic Analysis

For the samples of Spineli et al. [28], the *CYP1A2* gene extraction procedure followed the protocol proposed by Cornelis et al. [33], and for *ADORA2A*, that by Deckert et al. [34]. For the samples of Barreto et al. (unpublished), the *CYP1A2* gene extraction procedure followed the protocol proposed by Salinero et al. [35], and for *ADORA2A*, that by Muñoz et al. [18]. Genotyping for *CYP1A2* (rs762551) was successful in all participants. However, five participants were not successful in determining the polymorphism of the *ADORA2A* gene (rs5751876). The researchers and participants were blinded to their genetic polymorphisms until all statistical analyses had been completed. 

### 2.4. Statistical Analysis

Descriptives analysis are presented through frequencies and percentages for categorical variables and mean and standard deviation for numerical variables (CaffEQ-BR scores). The CaffEQ-BR concordance between full and brief versions was verified using the intraclass correlation coefficient (ICC). The type of ICC adopted was the absolute agreement considering the average agreement of the two applications. The ICC calculation was based on a two-way mixed model. According to Cicchetti [36], an excellent ICC agreement was considered when ≥0.75 was found between the two responses. The ability of CaffEQ-BR to identify the presence (or absence) of the *CPY1A2* and *ADORA2A* gene polymorphisms was evaluated by the receiver operating characteristic (ROC) curve. The area under the ROC curve (AUC) ranged from 0 to 1, an AUC = 0.5 indicates that CaffEQ-BR has no discrimination capability, and AUC = 0 or 1 corresponds to perfect discrimination. In addition, AUC 0.0–0.5 or 0.5–1.0 indicates that lower/higher CaffEQ-BR values indicate evidence of a positive state [37]. The analyses were performed with two-tier sample clusters using C allele carriers and homozygous AA genotype for *CYP1A2* and TT homozygotes for *ADORA2A*, and three-tier clusters using AA, AC, and CC genotypes for *CYP1A2* and TT, CT, and CC genotypes for *ADORA2A*. The ICC and AUC estimates are presented with their respective 95% confidence intervals and were evaluated by IBM SPSS Software version 22 (IBM SPSS Statistics for Windows, IBM Corp, Armonk, NY, USA).

## 3. Results

### 3.1. Sample Profile

This study was conducted with 71 Brazilian adults who were habitual caffeine consumers (244 ± 161 mg·day^−1^). The sample profile consisted of 90.1% males, 25 ± 8 y, body mass index 23.7 ± 3.9 kg/m^2^ (69.0% eutrophic [38]), 33.8% completed high school, and 83.1% presented an average monthly family income above one minimum wage (R$ 1212 = US$ 246.63) (May 2022) (Table 1). 

### 3.2. CYP1A2 and ADORA2A Genotypes

The frequency of AA homozygotes for the *CYP1A2* gene was 47.9% (*n* = 34), and that of C allele carriers (AC and CC genotypes) was 52.1% (*n* = 37). For the *ADORA2A* gene, 22.7% (*n* = 15) were TT homozygotes and 77.3% (*n* = 51) were C allele carriers (TC and CC genotypes).

### 3.3. CaffEQ-BR Full and Brief Questionnaires Agreement

All factors showed excellent agreement (ICC > 0.75; Table 2), except item 6: “Anxiety/negative physical effects” (ICC = 0.6). The scores obtained for the full and brief CaffEQ-BR showed an ICC agreement between the two versions (full with 47 items and brief with 21 items) (Table 2). 

### 3.4. CaffEQ-BR Discriminatory Capacity for CYP1A2 and ADORA2A Genotypes

The analyses performed with two-tier clusters (C allele carriers and homozygous carriers—AA for *CYP1A2* and TT for *ADORA2A*) obtained a better discriminatory capacity than with three-tier clusters (AA, AC, and CC for CYP1A2; and TT, CT, and CC for ADORA2A) (data not presented).

Table 3 shows the AUC results for CaffEQ-BR full and brief versions for the *CYP1A2* gene with the AA homozygotes as the reference level. Note that all seven factors and the overall result presented values near to 0.5, which indicates no discriminatory capacity for the *CYP1A2* genotypes. Figure 1 shows the CaffEQ-BR factor lines do not deviate from the reference diagonal line when compared to the pattern of responses recorded by genotype AA for CYP1A2, used as a reference for fast caffeine metabolism.

Table 4 shows the AUC results for the CaffEQ-BR full and brief versions for the *ADORA2A* gene with the TT homozygote group as the reference level. Note that of all seven factors, only factor 6 (anxiety/negative physical effects) showed discriminatory ability for the *ADORA2A* genotypes in the brief questionnaire (AUC = 0.293). In Figure 2, only factor 6 for the brief CaffEQ-BR is far from the reference diagonal line, which indicates its discriminatory capacity for *ADORA2A* genotypes.

## 4. Discussion

This is the first study that evaluated if an instrument (full and brief CaffEQ-BR questionnaires) can differentiate the polymorphisms of the *CYP1A2* and *ADORA2A* genes. Data showed that the full and brief versions of CaffEQ-BR were not able to differentiate the *CYP1A2* or *ADORA2A* genotypes in these adult Brazilian athletes. Of all factors, only factor 6 (anxiety/negative physical effects) in the brief questionnaire showed any discriminatory capacity for the genotype TT for the *ADORA2A* gene. However, the sample size was limited to two individuals, which precluded generalizing this result to an external population. Specifically, factor 6 in the brief CaffEQ-BR is composed of three questions related to the expected anxiety and negative physical effects of caffeine: “I don’t like the way I feel after drinking caffeine/coffee” (portuguese: *Não gosto de como me sinto depois de tomar cafeína /café*). “When I drink caffeine/coffee I get nervous” (portuguese: *Quando bebo cafeína /café fico nervoso*). “Caffeine/coffee makes me irritable” (portuguese: *Cafeína /café me deixa irritado*). Anxiety and sleep disturbance (factors 6 and 7) have been associated with the *ADORA2A* gene [27]; therefore, future studies should consider these factors when developing a questionnaire capable of discriminating between *ADORA2A* polymorphisms.

The CaffEQ-BR questionnaires were not capable of discriminating between genotypes for *CYP1A2*, suggesting differences in caffeine metabolism do not lead to different expectations regarding caffeine effects. The importance of caffeine metabolism for exercise performance following supplementation is still unclear and inconsistent [12,14]. Most studies have shown no influence of *CYP1A2* genotype on caffeine’s ergogenic effect [18,35,40,41], though some studies do suggest that those with slower metabolism (i.e., C-allele carriers) might have less benefit [15,17,42]. No study has shown differential effects of caffeine on physiological measurements or side-effects between *CYP1A2* genotypes. Differences in caffeine metabolism may not sufficiently alter variables that would modify caffeine expectancy, as the questionnaires here may provide questions that are too vague and generalized to detect differences in *CYP1A2* genotypes. It cannot be ruled out that other questionnaires related to how symptoms develop and persist over time following caffeine ingestion may provide a more accurate method of determining an individual’s caffeine metabolism genotype.

Studies that aim to find differences in the polymorphisms of the *ADORA2A* gene seem more consistent, especially regarding the negative effects of caffeine in more sensitive individuals (TT homozygotes), such as increased anxiety and sleep disturbance at higher doses (>6 mg·day^−1^) [10]. In the articles published by the present research group [25,26] and in all studies using the CaffEQ [24,43,44], the Likert scale scores were low (<3) in factor 6 (anxiety/negative physical effects). Huntley and Juliano [24] showed that daily consumers of caffeine presented high scores (≥4) in factors 1, 2, 3, 4, and 5 (related to expectation of dependence and beneficial effects of caffeine). However, irregular consumers of caffeine presented high scores (≥4) for factors 6 and 7 (related to negative effects such as anxiety and sleep disturbance). Studies suggest that individuals who may experience more adverse effects from caffeine, especially with dosages exceeding the safe limit (> 6 mg·day^−1^), are likely TT homozygotes for the *ADORA2A* gene [27,45]. Thus, it was surprising, and contrary to our hypothesis, that the questionnaires used here could not differentiate between *ADORA2A* genotypes.

Individuals with high dependency scores present strong correlation and high scores in CaffEQ factors 1, 2, 3, 4, and 5 (related to expectation of dependence and beneficial effects). However, individuals who reported a desire to reduce or eliminate caffeine consumption from their routines had high scores on factors 6 and 7 (related to anxiety/negative physical effects and sleep disturbance) [24]. In the study by Kearns et al. [44], factor 6 was associated with other validated questionnaires about anxiety, appetite suppression, and sleep disturbance. Schott et al. [43], who validated the CaffEQ for German-speaking countries, also found a negative correlation between mean consumption of caffeine and negative symptoms. This reinforces the hypothesis that habitual caffeine consumers are usually people with the genetic profiles to experience favorable effects from caffeine intake, and consequently greater chances of dependence. In addition, people who are very sensitive to caffeine may experience more adverse effects and avoid its consumption.

In all versions of the CaffEQ, the questionnaire presents more factors to support people who experience beneficial effects or a possible dependence on caffeine than negative effects, such as anxiety and sleep disturbance [24,25,26,43,44]. Therefore, the instrument to discriminate the individuals’ genetic variations regarding *ADORA2A* needs to be calibrated to have high sensitivity and specificity for those individuals who experience negative effects from caffeine. The present results suggest that future studies should include more individuals with low/irregular caffeine intake, as these may be, to a large extent, of the TT genotype for the *ADORA2A* gene. 

A recent caffeine expectancy questionnaire (CaffCo) stressed the methodological caution needed to balance the number of positive/negative caffeine effects factors in the development of this kind of questionnaire [46]. The CaffCo may be an alternative questionnaire to be tested in the future for the capacity to discriminate the polymorphisms for *CYP1A2* and *ADORA2A* genes. Furthermore, a good alternative could be to carry out an exploratory factor analysis, followed by a confirmatory analysis based on a biobank of individuals genotyped for *CYP1A2* and *ADORA2A* polymorphisms in an attempt to assess the ability to differentiate the genotypes according to the responses to the caffeine expectancy questionnaire. 

The current data suggest that the CaffEQ-BR questionnaire cannot be used to differentiate individuals for the *CYP1A2* and *ADORA2A* genotypes. Nonetheless, regarding the practical applications, the CaffEQ-BR remains a useful tool to understand the expectation of caffeine intake to assess any potential risk of bias in clinical trials in sports science due to the possible ergogenic effects of placebo associated with the expected effects of caffeine in the placebo/control group [47,48], since higher or lower expectations about the effect of caffeine may alter outcomes measures [12,14].

The strengths of the present study are that we explored the capacity of both full and brief versions of the CaffEQ-BR questionnaires to discriminate between *CYP1A2* and *ADORA2A* genotypes; importantly, we also showed good agreement between both versions of the questionnaire, confirming our previous work [26]. Some potential limitations include the small sample size of homozygous participants (CC for *CYP1A2* and TT for *ADORA2A*) and the small number of female participants. However, it is unclear if the questionnaire could lead to different discriminatory capacities for genotypes between men and women. Nevertheless, we recommended that future studies balance the sample profile in terms of gender. The research was conducted with the participants previously enrolled in the study of Spineli et al. [28] and Barreto et al. (unpublished), who performed the genotype analysis of their athletes. Furthermore, considering the COVID-19 pandemic period during the study conduction, it was not possible to increase our sample size or female representativeness. 

The analyses were even less discriminatory when performed with three-tier clusters (AA, AC, and CC for *CYP1A2*; and TT, CT, and CC for *ADORA2A*), due to high sample segmentation and analysis with small groups. To achieve a more expressive number of genotypes of CC and TT for the *CYP1A2* and *ADORA2A* genes (respectively), future studies should enroll larger samples due to the small presence of these homozygous individuals in the general population. We also encourage further studies to evaluate other applications of the CaffEQ-BR with more specific purposes, such as its application to the effect of placebo-controlled caffeine supplementation in clinical trials, as performed by Shabir et al. [49], with heart rate variability monitorization [50] and salivary paraxanthine level [51] as control covariates.

## 5. Conclusions

The CaffEQ-BR (full and brief versions) was not able to differentiate genotypes for the *CYP1A2* and *ADORA2A* genes in this healthy adult Brazilian athlete population. Future studies should replicate this research in a large sample, and include low caffeine consumers as a control group, thereby calibrating the caffeine expectancy questionnaire to focus on aspects of anxiety and increased negative effects in search of discriminating the TT genotype for the *ADORA2A* gene.

## Figures and Tables

**Figure 1 nutrients-14-03355-f001:**
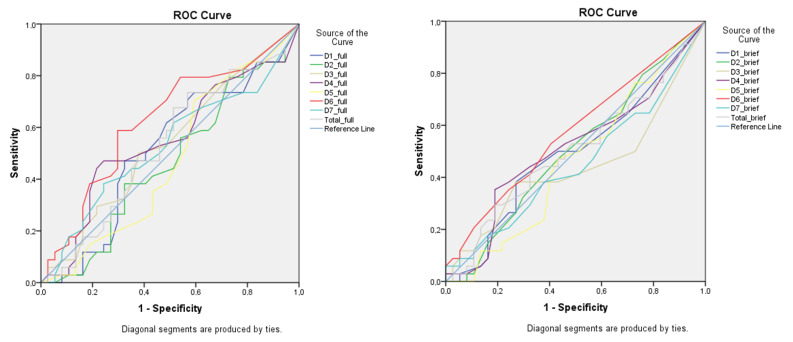
ROC curve of the CaffEQ-BR full (**left**) and brief (**right**) versions. The colored lines are the recorded scores of the 7 CaffEQ-BR factors for discrimination of the *CYP1A2* genotypes considering AA group as the reference level (0.5 = reference diagonal line, no discriminatory for sensitivity or specificity). Factors: D1: withdrawal/dependence; D2: energy/work enhancement; D3: appetite suppression; D4: social/mood enhancement; D5: physical performance enhancement; D6: anxiety/negative physical effects; D7: sleep disturbance.

**Figure 2 nutrients-14-03355-f002:**
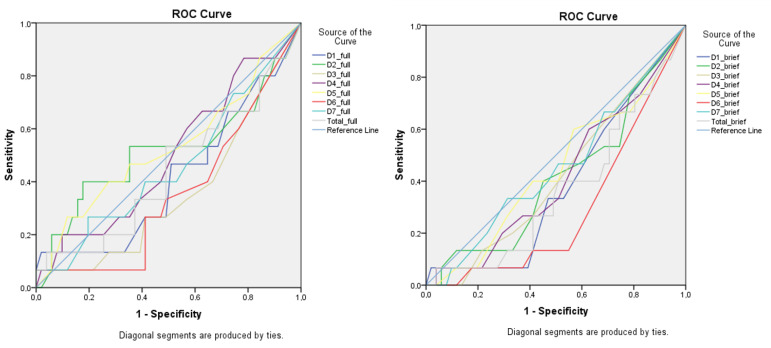
ROC curves of the CaffEQ-BR full (left) and brief (right) versions. The colored lines are the recorded scores of the 7 CaffEQ-BR factors for discrimination of the *ADORA2A* genotypes considering TT group as the reference level (0.5 = reference diagonal line, no discriminatory for sensitivity or specificity). Factors: D1: withdrawal/dependence; D2: energy/work enhancement; D3: appetite suppression; D4: social/mood enhancement; D5: physical performance enhancement; D6: anxiety/negative physical effects; D7: sleep disturbance.

**Table 1 nutrients-14-03355-t001:** Sociodemographic data and sample profile (*n* = 71).

	Categories	Total (*N* = 71)	
		*n*	%
Sex	Male	64	90.1
	Female	7	9.9
Age	<30 years	55	77.5
	≥30 years	16	22.5
Body Mass Index (kg/m^2^) *	<18.5	2	2.8
	18.5–24.9	49	69.0
	25–29.9	14	19.7
	≥30	6	8.5
Self-identification	Asia descendants	5	7.0
	White	24	33.8
	Indigenous	3	4.3
	Pardo	29	40.8
	Black	9	12.7
	Without description	1	1.4
Physical Exercises (≧150 min/week) **	No	26	36.6
	Yes	45	63.4
Degree of Education	Complete elementary school	1	1.4
	Incomplete high school	9	12.7
	Complete high school	24	33.8
	Incomplete graduated	22	30.0
	Graduated	6	8.5
	Postgraduate studies	9	12.7
Income (BRL) ***	Up to 1000.00	12	16.9
	1000.01 to 2000.00	18	25.4
	2000.01 to 3000.00	17	24,0
	3000.01 to 5000.00	15	21.1
	5000.01 to 10,000.00	5	7.0
	Above 10,000.00	4	5.6
Self-reported chronic diseases	No	67	94.4
	Yes	4	5.6

Note: * Body mass index (BMI) followed the criteria adopted by the World Health Organization (WHO) [38]: underweight (BMI < 18.5kg/m^2^), adequate (BMI between 18.5 and 24.9kg/m^2^), overweight (BMI between 25 and 29.9kg/m^2^), and obesity (BMI ≥ 30kg/m^2^); ** The cutoff point according to the WHO [39] was adopted, with a minimum workload that indicates whether the participant was physically active at the time of participating in the research; *** 5.00 BRL = 1.00 USD on May 2022.

**Table 2 nutrients-14-03355-t002:** Means (DP) and intra-class correlation coefficients (ICC) between scores of the CaffEQ-BR full and brief versions (*n* = 71).

Factors	Full	Brief	ICC * (CI 95%)
1. Withdrawal/dependence	2.33 (1.24)	2.06 (1.22)	0.851 (0.754–0.909)
2. Energy/work enhancement	3.17 (1.43)	3.13 (1.41)	0.879 (0.806–0.924)
3. Appetite suppression	1.87 (0.97)	1.70 (0.93)	0.769 (0.631–0.856)
4. Social/mood enhancement	2.62 (1.29)	2.55 (1.34)	0.907 (0.850–0.942)
5. Physical performance enhancement	3.16 (1.51)	2.94 (1.46)	0.891 (0.824–0.932)
6. Anxiety/negative physical effects	1.84 (0.89)	1.53 (0.80)	0.600 (0.356–0.751)
7. Sleep disturbance	2.58 (1.41)	2.49 (1.38)	0.858 (0.772–0.911)
Overall	2.44 (0.98)	2.34 (0.90)	0.856 (0.777–0.910)

* An excellent ICC agreement is considered when ≥ 0.75 was found between the two responses.

**Table 3 nutrients-14-03355-t003:** Area under the ROC curve (AUC) of CaffEQ-BR full and brief versions for *CYP1A2* genotypes.

Factors	AUC * (CI 95%)	
	Full	Brief
1. Withdrawal/dependence	0.513 (0.376–0.651)	0.496 (0.360–0.632)
2. Energy/work enhancement	0.464 (0.329–0.599)	0.504 (0.369–0.640)
3. Appetite suppression	0.537 (0.402–0.672)	0.443 (0.304–0.582)
4. Social/mood enhancement	0.548 (0.411–0.686)	0.514 (0.376–0.651)
5. Physical performance enhancement	0.472 (0.336–0.608)	0.467 (0.332–0.602)
6. Anxiety/negative physical effects	0.624 (0.492–0.757)	0.575 (0.441–0.709)
7. Sleep disturbance	0.528 (0.391–0.665)	0.453 (0.318–0.589)
Overall	0.529 (0.393–0.665)	0.504 (0.367–0.640)

* Genotype AA group is the reference level (0.5 = no discriminatory effects).

**Table 4 nutrients-14-03355-t004:** Area under the ROC curve (AUC) of CaffEQ-BR full and brief versions for *ADORA2A* genotypes.

Factors	AUC * (CI 95%)	
	Full	Brief
1. Withdrawal/dependence	0.415 (0.245–0.585)	0.385 (0.233–0.537)
2. Energy/work enhancement	0.516 (0.324–0.708)	0.414 (0.250–0.578)
3. Appetite suppression	0.358 (0.199–0.516)	0.422 (0.267–0.577)
4. Social/mood enhancement	0.498 (0.330–0.666)	0.417 (0.257–0.577)
5. Physical performance enhancement	0.529 (0.350–0.709)	0.444 (0.279–0.609)
6. Anxiety/negative physical effects	0.356 (0.201–0.510)	0.293 (0.155–0.431)
7. Sleep disturbance	0.461 (0.290–0.632)	0.455 (0.289–0.621)
Overall	0.443 (0.271–0.616)	0.367 (0.211–0.522)

* Genotype TT group is the reference level (0.5 = no discriminatory effects).

## Data Availability

Not applicable.

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
