# Peer review of "Can the Brazilian Caffeine Expectancy Questionnaires Differentiate the CYP1A2 and ADORA2A Gene Polymorphisms?—An Exploratory Study with Brazilian Athletes"

_nutrients, 2022, doi:10.3390/nu14163355_

Round 1

Reviewer 1 Report

Dear authors,

It was a pleasure to read this interesting cross-sectional study in which Caffeine questionnaires was used to differentiate the polymorphisms of genes responsible for caffeine metabolism in liver. This manuscript is well-written and informative. I have some questions which are listed below.

1.     In page 3 , “In the Spineli et al. study [23], the participants were healthy athletes engaged in volleyball, athletics, ..... “ What do you mean by athletes? please clarify.

2.     In Table 1, you have mentioned the duration of physical activity (150 min/week). Do you have any data on the intensity of the physical activity to report?

3.     Would have you expected different results, had you included more females in this project? Please provide some context in this respect in the discussion.

4.      What are the exact practical applications of this research? I would suggest adding a line to expand on this.

Author Response

Can the Brazilian Caffeine Expectancy Questionnaires differentiate the CYP1A2 and ADORA2A gene polymorphisms? – An exploratory study with Brazilian athletes

ID: Nutrients-1847249

Dear reviewer,

Thank you for the opportunity to improve our manuscript.

  1. In page 3 , “In the Spineli et al. study [23], the participants were healthy athletes engaged in volleyball, athletics, ..... “What do you mean by athletes? please clarify.

A: The criteria of Pauw et al. (2013) and McKay et al. (2021) were applied for athletes' classification of the study by Barreto et al. and Spineli et al.,. This information has now been included (lines 137-142).

McKay AK, Stellingwerff T, Smith ES, et al. Defining Training and Performance Caliber: A Participant Classification Framework. International Journal of Sports Physiology and Performance. 2022;17(2):317-331.

Pauw KD, Roelands B, Cheung SS, de Geus B, Rietjens G, Meeusen R. Guidelines to Classify Subject Groups in Sport-Science Research. International Journal of Sports Physiology and Performance. 2013;8(2):111-122.

  1. In Table 1, you have mentioned the duration of physical activity (150 min/week). Do you have any data on the intensity of the physical activity to report?

A: We appreciate the observation. The cut-off point adopted was according to WHO (2018) recommendations as a minimum load of exercise to indicate physically active status. Unfortunately, no information regarding intensity was collected. However, the online form specified that physical activity was activity with specific objectives of gaining physical capabilities with a training program. Therefore, activities of daily living were not counted as physical exercise according to WHO. A footnote has been included in the table 1.

World Health Organization. Global action plan on physical activity 2018-2030: more active people for a healthier world. World Health Organization, 2018.

  1. Would you have expected different results, had you included more females in this project? Please provide some context in this respect in the discussion.

A: It is unclear if the questionnaire could lead to different discriminatory capacities for genotypes between males and females. The small number of female participants was included as a limitation at the end of the discussion section (lines 406-412).

  1. What are the exact practical applications of this research? I would suggest adding a line to expand on this.

A: The current data suggest that the CaffEQ-BR questionnaire can not be used to differentiate individuals for the CYP1A2 and ADORA2A genotypes. Nonetheless, the CaffEQ-BR remains a useful tool to understand the expectation of caffeine intake to assess any potential risk of bias in clinical trials in sports science due to the possible ergogenic effects of placebo associated with the expected effects of caffeine in the placebo/control group (Beedie et al., 2006; Saunders et al., 2017), since higher or lower expectations about the effect of caffeine may alter outcomes measures.

This information has now been included in the lines (391-398) in the discussion section. Thanks for your contribution to improve our maniscript.

Beedie CJ, et al. Placebo effect of caffeine in cycling performance. Med Sci Sports Exerc. 2006;38:2159–2164.

Saunders B, et al. Placebo in sports nutrition: a proof-of-principle study involving caffeine supplementation. Scand J Med Sci Sports 2017;27(11):1240-1247.

Reviewer 2 Report

People respond differently to the psychoactive substance caffeine. The differences in response are partly due to single nucleotide polymorphisms in the CYP1A2 and ADORA2A gene. Currently, there is no validated instrument to predict genetic variations in these two genes. This study tests the ability of the Brazilian Caffeine Expectancy Questionnaires to differentiate polymorphisms in these two genes based on an individual’s perceived effects of caffeine.

The paper flows well, and the data is appropriately presented. However, I have a few minor comments.

Table 3 should be on one page, not broken between two pages.

The written text in Figures 1 and 2, for example, x and y axis labels, is difficult to read. All the text on the figures should be larger to make it readable.

Discussion, first paragraph: The word questionary, the last word of the paragraph, is not a word.

Discussion, 7th paragraph: The sentence that starts “This reinforces…” is missing some indefinite articles.

Conclusion, last full line: Something is missing, “…negative effects in search of discriminate the TT…”

Author Response

Caro revisor,

Obrigado pela oportunidade de melhorar nosso manuscrito!

Abstrato

  • Mantenha as palavras-chave entre 3-5 palavras.

R: Alterado de acordo.

Introdução

  • Mais referências são necessárias para as seguintes frases:

Para melhorar o desempenho esportivo, a dose recomendada varia de 3 a 6 mg.massa corporal-1 [4,5]. Para melhorar os aspectos cognitivos e a vigília, 1 a 3 mg. massa corporal-1 é uma dose eficaz [6,7].

R: Agora incluímos mais algumas referências, conforme sugerido. Consulte as linhas 47-49.

  1. Por favor, especifique o tipo de desempenho esportivo na seguinte frase:

Variantes genéticas podem afetar o metabolismo da cafeína [10] induzindo diferentes efeitos da cafeína em relação à percepção de esforço, fadiga, sono, apetite e efeitos adversos ou benéficos, por exemplo, no desempenho esportivo [11,12].

A: We appreciate the consideration. The type of sports was inserted in lines 56-60.

  1. 3-5 references are needed for the following sentences:

In nutrigenomic studies, caffeine is one of the most studied substance in clinical trials assessing the interaction between gene polymorphisms and sports performance [13].

A: We appreciate the considerations. Three additional references were included  (lines 60-63).

  1. The following sentence needs to be rewritten:

In clinical trials, is pivotal to understand the sample profile and is desirable to predict the expected effects of caffeine according to the genotyping of the participant to minimize this risk of bias, mainly in studies with a small sample [6].

R: Agradecemos sua observação, reescrevemos a frase para deixar claro para os leitores (linhas 78-85).

  1. A hipótese do estudo precisa ser adicionada ao final da “Introdução”.

R: A hipótese foi adicionada no final da seção de introdução (linhas 104-110).

  1. Discussão

  • A seção “Discussão” precisa esclarecer a novidade do estudo.

R: A novidade do estudo foi incluída no início da seção de discussão (linhas 292-294).

  1. A parte das limitações do estudo está faltando no manuscrito, e existem inúmeras limitações a serem listadas.

R: As limitações são apresentadas no final da seção de discussão (linhas 405-427).

Reviewer 3 Report

The study is informative and adds value to the current research body; however, some points should be addressed.

Abstract

·         Keep the keywords between 3-5 words.

Introduction

·         More references are needed for the following sentences:

To improve sports performance, the recommended dose varies from 3 to 6 mg.body mass-1 [4,5]. To improve cognitive aspects and wakefulness, 1 to 3 mg. body mass-1 is an effective dose [6,7].

·         Please specify the type of sports performance in the following sentence:

Genetic variants can affect caffeine metabolism [10] inducing different effects of caffeine regarding the perception of effort, fatigue, sleep, appetite, and adverse or beneficial effects, e.g., on sports performance [11,12].

·         3-5 references are needed for the following sentences:

In nutrigenomic studies, caffeine is one of the most studied substance in clinical trials assessing the interaction between gene polymorphisms and sports performance [13].

·         The following sentence needs to be rewritten:

In clinical trials, is pivotal to understand the sample profile and is desirable to predict the expected effects of caffeine according to the genotyping of the participant to minimize this risk of bias, mainly in studies with a small sample [6].

·         The study hypothesis needs to be added at the end of the “Introduction”.

Methods

·         Day-to-day test reliability, CV range, and intraclass correlation coefficients for the assessments must be included for ALL the assessments.

Discussion

·         The “Discussion” section needs to clarify the study's novelty.

·         The Study limitations part is missing from the manuscript, and there are numerous limitations to be listed.

Author Response

Dear reviewer,

Thank you for the opportunity to improve our manuscript!

Abstract

  • Keep the keywords between 3-5 words.

A: Changed accordingly.

Introduction

  • More references are needed for the following sentences:

To improve sports performance, the recommended dose varies from 3 to 6 mg.body mass-1 [4,5]. To improve cognitive aspects and wakefulness, 1 to 3 mg. body mass-1 is an effective dose [6,7].

A: We have now included some more references, as suggested. Please see lines 47-49.

  • Please specify the type of sports performance in the following sentence:

Genetic variants can affect caffeine metabolism [10] inducing different effects of caffeine regarding the perception of effort, fatigue, sleep, appetite, and adverse or beneficial effects, e.g., on sports performance [11,12].

A: We appreciate the consideration. The type of sports was inserted in lines 56-60.

  • 3-5 references are needed for the following sentences:

In nutrigenomic studies, caffeine is one of the most studied substance in clinical trials assessing the interaction between gene polymorphisms and sports performance [13].

A: We appreciate the considerations. Three additional references were included  (lines 60-63).

  • The following sentence needs to be rewritten:

In clinical trials, is pivotal to understand the sample profile and is desirable to predict the expected effects of caffeine according to the genotyping of the participant to minimize this risk of bias, mainly in studies with a small sample [6].

A: We appreciate your observation, we have rewritten the sentence to make it clear to the readers (lines 78-85).

  • The study hypothesis needs to be added at the end of the “Introduction”.

A: The hypothesis was added at the end of the introduction section (lines 104-110).

Methods

  • Day-to-day test reliability, CV range, and intraclass correlation coefficients for the assessments must be included for ALL the assessments.

A: We appreciate the considerations. We have previously shown the full (all ICCs > 0.9; Mendes et al., 2020; https://doi.org/10.3390/nu12082248) and brief ((all ICCs > 0.9; Mendes et al., 2021; https://doi.org/10.3389/fnut.2021.695385) versions of the questionnaires have excellent reliability in the Brazilian population. This information has now been included in the manuscript (lines 158-161).

Discussion

  • The “Discussion” section needs to clarify the study's novelty.

A: The study's novelty was included at the beginning of the discussion section (lines 292-294).

  • The Study limitations part is missing from the manuscript, and there are numerous limitations to be listed.

A: The limitations are presented at the end of the discussion section (lines 405-427).
